# CRISPR/Cas12a-Based One-Tube RT-RAA Assay for PoRV Genotyping

**DOI:** 10.3390/ijms26146846

**Published:** 2025-07-16

**Authors:** Mingfang Bi, Zunbao Wang, Kaijie Li, Yuhe Ren, Dan Ma, Xiaobing Mo

**Affiliations:** 1College of Veterinary Medicine, State Key Laboratory for Diagnosis and Treatment of Severe Zoonotic Infectious Diseases, Key Laboratory for Zoonosis Research of the Ministry of Education, and Institute of Zoonosis, Jilin University, Changchun 130062, China; bi_mingfang@gibh.ac.cn (M.B.); zunbao22@mails.jlu.edu.cn (Z.W.); 17838408328@163.com (K.L.); renyh24@mails.jlu.edu.cn (Y.R.); madandan@jlu.edu.cn (D.M.); 2CAS Key Laboratory of Regenerative Biology, Guangdong Provincial Key Laboratory of Stem Cell and Regenerative Medicine, Guangzhou Institutes of Biomedicine and Health, Chinese Academy of Sciences, Guangzhou 510530, China

**Keywords:** PoRV, RT-RAA, CRISPR/Cas12a, genotyping

## Abstract

Porcine rotavirus (PoRV), a primary etiological agent of viral diarrhea in piglets, frequently co-infects with other enteric pathogens, exacerbating disease severity and causing substantial economic losses. Its genetic recombination capability enables cross-species transmission potential, posing public health risks. Globally, twelve G genotypes and thirteen P genotypes have been identified, with G9, G5, G3, and G4 emerging as predominant circulating strains. The limited cross-protective immunity between genotypes compromises vaccine efficacy, necessitating genotype surveillance to guide vaccine development. While conventional molecular assays demonstrate sensitivity, they lack rapid genotyping capacity and face technical limitations. To address this, we developed a novel diagnostic platform integrating reverse transcription recombinase-aided amplification (RT-RAA) with CRISPR–Cas12a. This system employs universal primers for the simultaneous amplification of G4/G5/G9 genotypes in a single reaction, coupled with sequence-specific CRISPR recognition, achieving genotyping within 50 min at 37 °C with 10^0^ copies/μL sensitivity. Clinical validation showed a high concordance with reverse transcription quantitative polymerase chain reaction (RT-qPCR). This advancement provides an efficient tool for rapid viral genotyping, vaccine compatibility evaluation, and optimized epidemic control strategies.

## 1. Introduction

Porcine rotavirus, a leading etiological agent of viral diarrhea in piglets, is ubiquitously prevalent in global swine herds and frequently co-infects with enteric pathogens such as the porcine epidemic diarrhea virus (PEDV) [1], exacerbating clinical outcomes and causing substantial economic losses [2]. Notably, PoRV poses a potential public health threat due to its capacity for genetic recombination, enabling the emergence of novel strains with cross-species transmission risks to humans and other animals [3,4]. Currently, no specific antiviral therapies exist for rotavirus infection, and disease control primarily relies on vaccination [5,6]. However, the extensive genetic diversity of PoRV, with twelve G genotypes and thirteen P genotypes associated with porcine diarrhea, poses significant challenges [7]. Predominant circulating strains include the G9 genotypes, followed by G5, G3, and G4 genotypes [8]. The limited cross-immunoprotective efficacy among distinct genotypes compromises the comprehensive protective capacity of current vaccines [9,10], underscoring the critical need to monitor the genotype distribution of circulating strains to guide vaccine development and targeted interventions [11].

Innovations in isothermal nucleic acid amplification technologies for point-of-care diagnostics. The advancement of isothermal nucleic acid amplification technologies has driven the development of point-of-care (POC) diagnostic tools. Among these, RT-RAA provides an efficient solution for rapid RNA virus detection by integrating reverse transcription with recombinase-mediated amplification. Unlike conventional PCR, RT-RAA eliminates the need for complex thermal cycling equipment, requiring only a constant temperature (37–42 °C) to complete amplification. This significantly reduces testing costs and enhances field applicability. With advantages such as a rapid reaction time (20–30 min), high sensitivity, and operational simplicity, RT-RAA has emerged as a critical tool in pathogen diagnostics [12]. However, challenges persist in its application to on-site diagnostics due to false-positive signals caused by nonspecific amplification [13]. To address these limitations, CRISPR–Cas-based detection platforms have been developed and validated. The CRISPR–Cas system comprises CRISPR RNA (crRNA) and Cas proteins. For instance, the binding of Cas12a and crRNA forms a binary complex capable of sequence-specific recognition and the cleavage of target DNA. This interaction activates nonspecific trans-cleavage activity, enabling the degradation of single-stranded DNA reporters [14]. By coupling CRISPR–Cas with RT-RAA, the system eliminates nonspecific amplification signals from RT-RAA while RT-RAA enhances the sensitivity of CRISPR–Cas detection. This synergistic approach achieves both high specificity and ultrasensitive target identification, making it particularly suitable for low-resource settings.

Porcine rotavirus is a significant etiological agent responsible for diarrheal disease in piglets [15]. However, its diverse genotypes challenge conventional detection methods in its rapid differentiation of viral subtypes, thereby complicating vaccine selection and epidemic surveillance. Current diagnostic approaches include etiological identification [16,17], serological assays [18,19,20], and molecular biological techniques [21,22]. Advanced detection platforms now integrate molecular and immunological approaches, exemplified by TaqMan multiplex real-time quantitative PCR systems capable of the simultaneous identification of rotavirus alongside other enteric pathogens [1]. Complementing these nucleic acid-based methods, double-antibody sandwich quantitative ELISA has emerged as a robust serological tool for the specific quantification of porcine epidemic diarrhea virus (PEDV) infections, even in co-infection scenarios with rotavirus [23]. Most notably, the development of gold-magnetic nanoparticle-enhanced surface-enhanced Raman scattering (SERS) immunochromatographic assays represents a paradigm shift, enabling a rapid on-site differential diagnosis of PEDV and porcine rotavirus (PoRV) co-infections through the synergistic integration of nanomaterial engineering and spectroscopic detection modalities [15]. Although molecular detection technologies demonstrate advantages in sensitivity and specificity, they suffer from operational complexity, high costs, and a susceptibility to contamination [24]. With the evolution of CRISPR–Cas system mechanisms, its application potential in viral detection continues to expand [25,26]. CRISPR/Cas12a-based detection platforms, while sensitive, only confirm a viral presence without achieving precise genotyping [27]. To address this limitation, this study developed a novel diagnostic strategy integrating RT-RAA with the CRISPR/Cas12a system [28]. By designing “universal primers” to simultaneously amplify three genotypes (G4, G5, and G9) in a single reaction tube, primer interference was effectively eliminated. Leveraging the sequence-specific recognition capability of the CRISPR system, this method enables the differentiation of these genotypes within 50 min at 37 °C, exhibiting exceptional sensitivity (detection limit: 10^0^ copies/μL) (Figure 1). Clinical validation demonstrated a high consistency with conventional RT-qPCR results. This technological breakthrough provides a practical tool for rapid viral genotyping in swine farms and vaccine compatibility evaluation, significantly contributing to PoRV transmission control and the optimization of prevention strategies.

## 2. Results

### 2.1. Identification of Cas12a Protein Cleavage Activity

The purified Cas12a protein exhibited high purity and concentration following a series of purification steps (Appendix A). To characterize the cis- and trans-cleavage activities of the purified Cas12a protein, we conducted four experimental setups using pUC57-G4, pUC57-G5, and pUC57-G9 recombinant plasmids as templates. The nucleotide sequence alignment analysis of the different rotavirus subtypes revealed that the fragments amplified by RAA and primer sequences were conserved across almost all respective subtypes (Appendix A). Furthermore, the crRNA-targeted regions exhibited specificity for rotavirus genotypes G4, G5, and G9 (Figure 2). Reaction products were analyzed through 2% agarose gel electrophoresis. Figure 3a,c demonstrates the dsDNA cleavage activity of CRISPR/Cas12a systems (G4, G5, and G9) under varying conditions. The gel electrophoresis lanes are configured as follows: Lane 1—DNA ladder; Lane 2—target DNA alone; Lane 3—target DNA with Cas12a protein; Lane 4—target DNA with Cas12a protein and CRISPR RNA (crRNA); Lane 5—target DNA with crRNA alone. Complete cleavage efficiency of the target dsDNA was observed exclusively in Lane 4 (Cas12a + crRNA + target DNA), confirming that cis-cleavage activity required the presence of all three components (Cas12a, crRNA, and target DNA). Figure 3d–f validates the ssDNA trans-cleavage activity of the purified Cas12a protein using M13mp18 circular ssDNA. The lane configurations are Lane 1—DNA ladder; Lane 2—Cas12a protein + ssDNA + crRNA; Lane 3—ssDNA + target DNA + crRNA; Lane 4—Cas12a + ssDNA + crRNA + target DNA. The significant non-specific degradation of ssDNA occurred only in Lane 4, where the ternary Cas12a/crRNA/target DNA complex was formed. This result demonstrates that trans-cleavage activity is strictly dependent on the prior activation of Cas12a through target DNA recognition. The cis-cleavage of dsDNA targets by Cas12a systems occurs with full efficiency only when Cas12a, crRNA, and target DNA are co-present (Figure 3a−c); the trans-cleavage of ssDNA reporters is triggered exclusively upon ternary complex formation, as evidenced by the degradation of M13mp18 ssDNA in activated reactions (Figure 3d−f). These findings collectively demonstrate that the Cas12a protein possesses robust cis- and trans-cleavage capabilities.

### 2.2. Establishment and Optimization of RT-RAA System

Four primer pairs were initially designed for the RT-RAA of target genes (Table 1). An electrophoretic analysis (2% agarose gel) identified primer pair No. 4 as optimal, producing the highest yield of amplified products with minimal nonspecific bands (Appendix A). Then we optimized the RT-RAA reaction conditions from three aspects: reaction time, reaction temperature, and primer concentration. Reaction time: amplification products became detectable at 5 min, increased progressively until 20 min, and plateaued thereafter (Appendix A). A 20 min reaction time was selected for rapid diagnostics. Primer concentration: testing concentrations from 0.1 μM to 0.6 μM revealed maximal target amplification at 0.4 μM (Appendix A). Temperature: comparative analysis across 35–41 °C demonstrated slightly enhanced product yield at 37 °C (Appendix A).

### 2.3. Optimization of a Single-Tube RT-RAA–CRISPR/Cas12a Assay System

To minimize the risk of contamination, we developed a single-tube detection system. The RT-RAA reaction mixture was first added to the bottom of a centrifuge tube. Subsequently, 10 μL of the Cas12a detection system was loaded onto the inner surface of the tube lid [29]. After gently sealing the lid, RT-RAA was allowed to proceed to completion. Finally, a brief centrifugation step was performed to transfer the Cas12a system from the lid to the reaction mixture at the bottom of the tube, enabling Cas12a-mediated target cleavage [30,31,32]. To validate the feasibility of this method, template-containing positive controls and no-template negative controls were established for genes G4, G5, and G9. Fluorescence signals were quantitatively analyzed using a multifunctional microplate reader and visually confirmed with fluorescent detection tubes. As shown in the results (Appendix A), all the positive controls exhibited significant fluorescence, whereas no fluorescence was observed in the negative controls, confirming the effectiveness of the method.

Subsequently, to optimize the detection system, CutBuffer was selected as the reaction buffer based on the literature references [33], with Cas12a protein and crRNA concentrations fixed at 350 nM and 200 nM, respectively. For reaction time optimization (using G4 as the model target), the reaction progress was monitored at different time intervals via the microplate reader and fluorescent tubes. The results indicated that the reaction reached a plateau phase at 30 min (Figure 4a); thus, 30 min was chosen as the optimal cleavage duration to achieve rapid detection. To further optimize ssDNA concentration, a gradient series (125, 250, 500, 750, and 1000 nM) was tested under standardized conditions (37 °C, 30 min, excitation/emission wavelengths of 485/535 nm). Fluorescence signals at the reaction endpoint demonstrated that the 500 nM ssDNA concentration yielded the highest fluorescence increase in the positive groups and the maximal signal-to-noise ratio (S/N), establishing 500 nM as the optimal concentration (Figure 4b).

### 2.4. Sensitivity and Specificity of the Assay

To validate the specificity of the established RT-RAA–CRISPR/Cas12a detection method for G4, G5, and G9 genotypes, we tested representative strains including G4, G5, and G9, along with other viruses (PCV2, PCV3, AKAV, and EV71) under optimized reaction conditions. Fluorescence dynamics were monitored in real-time using both a multifunctional microplate reader and fluorescence visualization tubes during 30 min of incubation at 37 °C. The results demonstrated exclusive fluorescence enhancements in the G4, G5, and G9 groups (Figure 5). Specifically, an analysis of the microplate reader revealed significantly higher fluorescence values in these three groups compared to the negative controls, while other viral groups showed no statistical difference from the controls. Correspondingly, only the G4, G5, and G9 samples exhibited distinct green fluorescence signals under blue light excitation, confirming method specificity without cross-reactivity among the three genotypes.

For the sensitivity assessment, the recombinant plasmids pUC57-G4, pUC57-G5, and pUC57-G9 (10^4^–10^0^ copies/µL) were used as templates along with template-free negative controls. The optimized RT-RAA–CRISPR/Cas12a detection method was applied to detect the G4, G5, and G9 targets. The results analyzed via a multifunctional microplate reader and visual fluorescence tubes showed that the fluorescence signals for G4, G5, and G9 at 10^0^ copies/µL were detectable, with significant differences compared with negative controls. A green fluorescence at 10^0^ copies/µL was also visually observed under blue light. These findings indicate that the method achieves a sensitivity of 10^0^ copies/µL, demonstrating its high detection sensitivity.

### 2.5. Clinical Sample Testing

To evaluate the clinical applicability of the established RT-RAA–CRISPR/Cas12a detection system, we analyzed 31 clinical specimens using both a multifunctional microplate reader and fluorescence visualization tubes for signal interpretation (Figure 6). The detection rates for the G4, G5, and G9 genotypes were determined as 25.8% (7/31), 3.2% (1/31), and 48.4% (15/31), respectively. These findings were subsequently validated through parallel testing with a conventional RT-qPCR methodology (Appendix A). A comparative analysis revealed identical positivity counts for the G4 (7), G5 (1), and G9 (15) genotypes between the two detection platforms, demonstrating 100% concordance. Notably, the RT-RAA–CRISPR/Cas12a system maintained an equivalent diagnostic accuracy to qRT-PCR while exhibiting substantial improvements in its processing speed and operational efficiency.

## 3. Discussion

In the global livestock industry, porcine diarrhea-associated viral diseases contribute significantly to piglet mortality, with PoRV recognized as a leading causative agent [34]. PoRV, which belongs to the genus Rotavirus within the family Reoviridae, can be divided into three major groups: A, B, and C, among which group A exhibits the highest prevalence [35]. The viral outer capsid protein *VP7* gene serves as a key genetic marker. Based on the *VP7* gene, PoRV can be classified into 42G genotypes, with G4, G5, and G9 predominating in clinical isolates [36]. While the current nucleic acid detection methods for PoRV (PCR, qPCR, and digital PCR) are widely used [37], they suffer from critical limitations including operational complexity, reliance on expensive instrumentation, and the requirement for specialized training factors that render them impractical for rapid on-site applications.

First identified in the 1980s, the CRISPR system has now been adapted for diverse applications ranging from gene editing to nucleic acid detection [38]. Among CRISPR subtypes, the Type V CRISPR/Cas12a system offers distinct advantages: unlike other CRISPR systems, it functions without tracrRNA and specifically cleaves target DNA containing a protospacer adjacent motif (PAM) guided by crRNA [39,40]. Following the formation of a crRNA–DNA–Cas12a ternary complex, the activated Cas12a enzyme exhibits collateral ssDNA cleavage activity [41], a feature that enables visual detection through engineered probes [42]. Notably, CRISPR/Cas12a demonstrates concentration-dependent activity while high target concentrations ensure precise recognition through crRNA complementarity; low-abundance targets may trigger off-target effects, potentially leading to false-negative results [25]. To address this limitation, the pre-amplification of low-copy nucleic acids becomes essential. Conventional PCR/RT-PCR amplification, though effective, requires thermal cyclers and involves transfer steps that increase aerosol contamination risks. Isothermal alternatives like recombinase polymerase amplification (RAA) enable low-copy amplification without specialized equipment [43], yet remain prone to non-specific amplification. A synergistic combination of RAA and CRISPR/Cas12a effectively mitigates these individual shortcomings, as demonstrated in previous pathogen detection studies [44,45,46]. Building on this foundation, we developed an RT-RAA–CRISPR/Cas12a assay to discriminate between the epidemiologically dominant G4, G5, and G9 PoRV genotypes.

In this study, we established and validated a CRISPR/Cas12a-based method for the sensitive detection and differentiation of the PoRV G4, G5, and G9 genotypes. The RT-RAA–CRISPR/Cas12a assay demonstrated single-copy sensitivity for all three genotypes with a complete absence of cross-reactivity between the genotypes or with other common porcine viruses. Clinical validation using 31 field samples revealed detection rates of 25.8% (7/31) for G4, 3.2% (1/31) for G5, and 48.4% (15/31) for G9, showing full concordance with the RT-qPCR results. These findings establish this method as a robust platform for field-deployable rotavirus detection and genotyping, particularly suited for resource-limited settings.

## 4. Materials and Methods

### 4.1. Design of Target DNA, crRNA, Primers, and PUC57 Plasmid

To identify the conserved sequence regions for the G4, G5, and G9 genotypes that were suitable for RT-RAA, *VP7* gene fragments of rotavirus were retrieved from GenBank. Multiple sequence alignment was performed using MEGA 11.0 with the default parameters for local alignment. After identifying the conserved sequences, crRNAs were designed through the Benchling website (San Francisco, CA, USA; https://www.benchling.com/). Three crRNA segments were designed to target the conserved regions within each genotype while ensuring specificity across the genotypes. The *VP7* gene containing these conserved regions was ligated to the pUC57 plasmid (Sangon Biotech, Shanghai, China). Primer sequences for *VP7* pre-amplification were designed using the Oligo 7 software, resulting in four primer pairs (forward and reverse primers listed in Table 1). All primers, crRNAs, and the pUC57 plasmid were synthesized by Sangon Biotech Engineering Co., Ltd. (Shanghai, China).

### 4.2. Expression and Purification of pET-28b-LbCas12a Protein

The pET28b-Cas12a plasmid, stored previously, was used for protein expression in an *E. coli* prokaryotic expression system [33]. Protein expression was performed using an *E. coli* prokaryotic expression system, induced with 0.4 mM of isopropyl β-D-1-thiogalactopyranoside (IPTG). After 20 h, the bacterial pellets were resuspended in a lysis buffer (500 mM of NaCl, 20 mM of Tris-HCl (pH 7.9), 25 mM of potassium phosphate (pH 6.8), 10% glycerol, and 1 mM of DTT). The mixture was then lysed using a low-temperature high-pressure homogenizer (JNBIO, Guangzhou, China). Purification was carried out using Ni-NTA chromatography. The protein was dialyzed to 20 mM of Tris-HCl (pH 7.4) with 200 mM of NaCl and further purified by HiTrap Heparin HP. The final step utilized the ClearFirst-3000 purification system (Flash Chromatography, Suzhou, China). The purified LbCas12a protein was stored at −80 °C in a storage buffer (20 mM of Tris-HCl (pH 7.4) and 500 mM of NaCl) with 10% glycerol added to ensure stability.

### 4.3. Sample Collection and Viral RNA Extraction

Viral RNA was extracted from clinical samples using the YALEPIC^®^ Viral RNA Extraction Kit (YALI Biotech, China). Briefly, 200 µL of sample was mixed with 20 µL of proteinase K and 350 µL of YLPN lysis buffer, followed by purification through silica-membrane spin columns according to the manufacturer’s protocol.

### 4.4. Optimization of RT-RAA Conditions

Following the RAA operation protocol (https://www.twistdx.co.uk/support/, accessed on 9 November 2024), we incorporated a rehydration buffer (29.5 µL), primers, RNA, nuclease-free water, and a lyophilized pellet. To optimize the RT-RAA reaction conditions, four primer pairs were tested for amplification, and the most efficient pair was selected. Further optimization included adjusting the reaction temperature (35–41 °C), time (5–30 min), and primer concentration (100–600 nM). Amplified products were extracted with phenol/chloroform and analyzed by 2% agarose gel electrophoresis (120 V, 30 min).

### 4.5. Sensitivity and Specificity Verification of the RT-RAA–CRISPR/Cas12a Genotyping Detection Method

An ssDNA reporter molecule labeled with FAM and BHQ-1 was employed for fluorescence detection. Upon activation through the specific cleavage of PoRV genomic dsDNA, Cas12a subsequently cleaves the ssDNA reporter molecule via its nonspecific endonuclease activity. The reaction system comprised two compartments: the tube cap and tube bottom. The tube cap contained the RT-RAA components (primers, template RNA, and rehydration buffer), while the tube bottom contained optimized concentrations of 100 nM of Cas12a protein and 100 nM of crRNAs (G4, G5, and G9), 10 mM of MgCl_2_, 1 mM of DTT, 50 mM of Tris-HCl (pH 7.9), 100 mM of KCl, and 500 nM of ssDNA (F-Q) reporter molecule. Sensitivity was evaluated using serial dilutions of the target RNA for G4 (3.28 × 10^4^, 3.28 × 10^3^, 3.28 × 10^2^, 3.28 × 10^1^, 3.28 × 10^0^ copies/µL), G5 (3.28 × 10^4^, 3.28 × 10^3^, 3.28 × 10^2^, 3.28 × 10^1^, 3.28 × 10^0^ copies/µL), and G9 (3.28 × 10^4^, 3.28 × 10^3^, 3.28 × 10^2^, 3.28 × 10^1^, 3.28 × 10^0^ copies/µL). Specificity was assessed against non-target nucleic acids (PCV2, PCV3, AKAV, EV71, and two non-target genotypes). The fluorescence signals were quantified using a multifunctional microplate reader (λex: 485 ± 20 nm; λem: 528 ± 20 nm) or visualized with a blue-light transilluminator (λex: 470 nm).

### 4.6. Clinical Sample Validation

The RNA extracted from 31 porcine diarrhea or intestinal tissue samples was analyzed using both the established assay and a reference RT-qPCR method. The results from the two assays were compared to validate clinical performance.

## Figures and Tables

**Figure 1 ijms-26-06846-f001:**
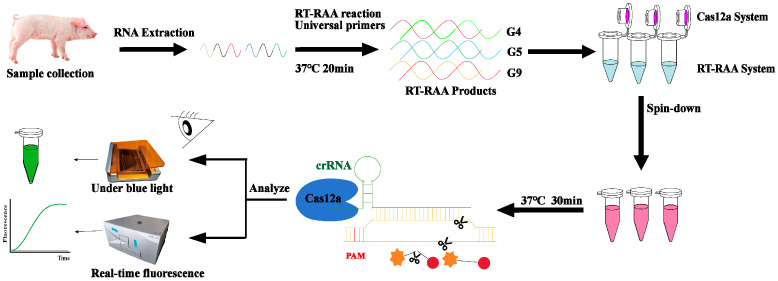
PoRV detection using RT-RAA–CRISPR/Cas12a one-tube assay. For detection of G4, G5, and G9 genotypes in PoRV, 10 μL of CRISPR/Cas12a reaction mixture was loaded into the cap of a centrifuge tube, while 9 μL of unreacted RT-RAA mixture and 1 μL of viral sample were added to the tube bottom. Following 20 min amplification at 37 °C, the CRISPR/Cas12a reaction in the tube cap bound to amplified nucleic acid targets in the tube bottom. Detection was performed using the RT-RAA–CRISPR/Cas12a one-tube assay, with results visualized through a fluorescent signal readout or direct fluorescent tube observation.

**Figure 2 ijms-26-06846-f002:**
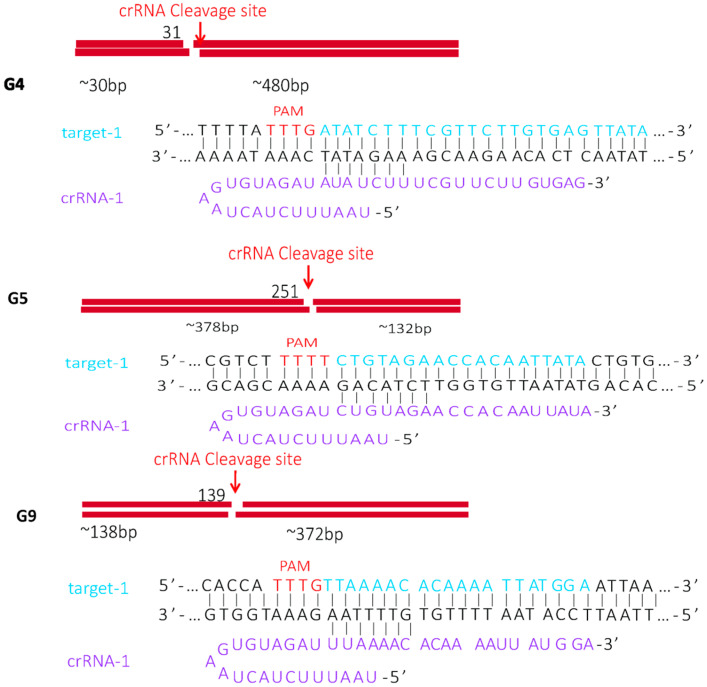
Analysis of recognition sequences and cleavage site. Analysis of PAM and recognition sequences. The PAM sequence is highlighted in red, while the target DNA and crRNA sequences are represented in blue and purple, respectively. RT-RAA product derived from the VP7 gene, including its cleavage site.

**Figure 3 ijms-26-06846-f003:**
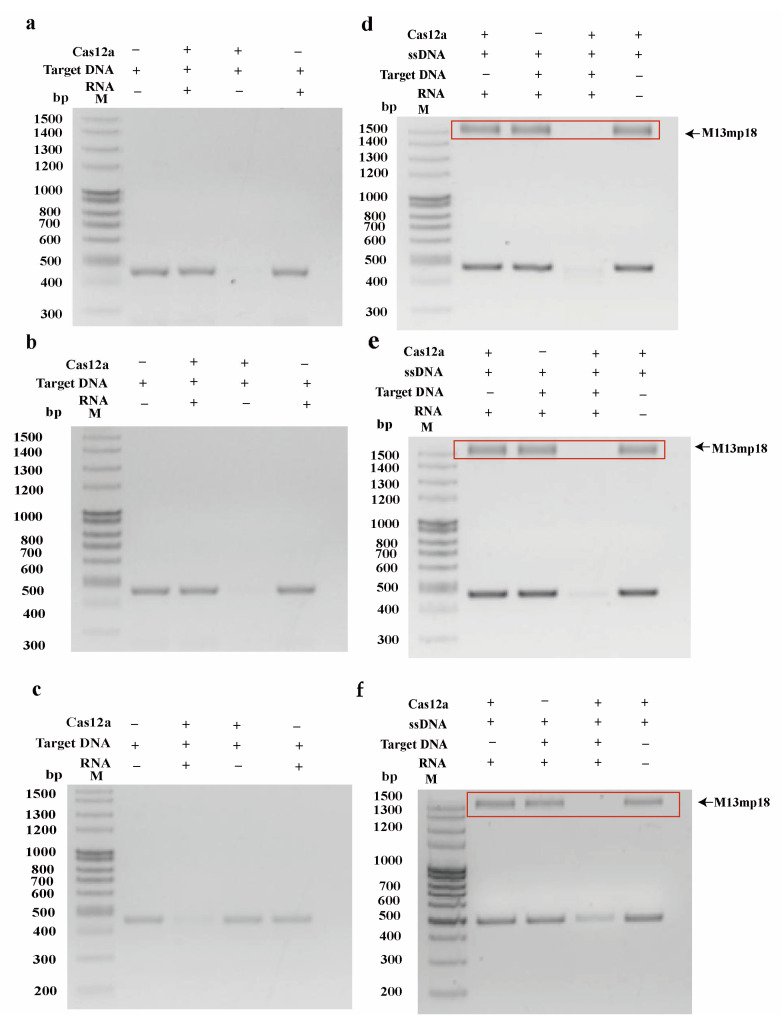
Validation of CRISPR/Cas12a cis-cutting and trans-cutting. In the cis-cleavage assay validating Cas12a activity, G4, G5, and G9 were specifically cleaved only when all reaction components were present. (**a**) Validation of G4 cis-cleavage activity. (**b**) Validation of G5 cis-cleavage activity. (**c**) Validation of G9 cis-cleavage activity. For trans-cleavage activity validation of Cas12a, M13mp18 circular single-stranded DNA (ssDNA) was introduced as a substrate to assess enzymatic function. Experimental results for G4, G5, and G9 demonstrated that activation of Cas12a by target nucleic acids resulted in ssDNA cleavage via trans-cleavage activity when all required reaction components were included. (**d**) Validation of G4 trans-cleavage activity. (**e**) Validation of G5 trans-cleavage activity. (**f**) Validation of G9 trans-cleavage activity.

**Figure 4 ijms-26-06846-f004:**
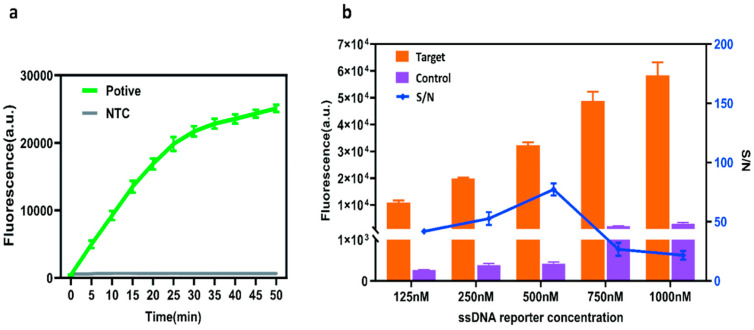
Optimized one-tube RT-RAA–CRISPR/Cas12a Assay System. (**a**) Optimization of reaction time for the final assay reaction. (**b**) Evaluation of ssDNA-FQ reporter concentration examines how varying levels affect the efficiency and sensitivity of the CRISPR/Cas12a assay, identifying the optimal concentration that maximizes signal amplification while minimizing background noise.

**Figure 5 ijms-26-06846-f005:**
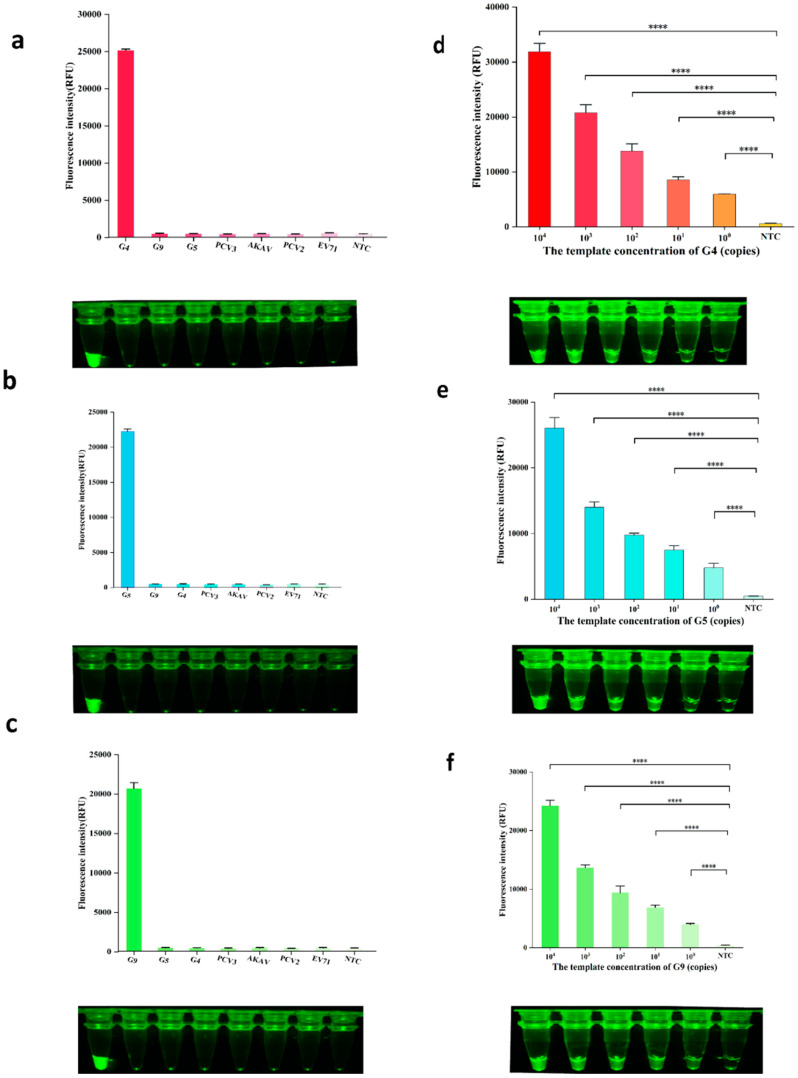
Specificity and sensitivity assessment. The optimized RT-RAA–CRISPR/Cas12a method was employed to validate the specificity and detect the sensitivity of G4, G5, and G9, respectively. Reaction results were visualized using a multifunctional zymography analyzer and fluorescent detection tubes. (**a**) Specificity validation of G4 using the established method. (**b**) Specificity validation of G5 using the established method. (**c**) Specificity validation of G9 using the established method. (**d**) Detection limit determination for G4 with nucleic acid copy numbers ranging from 10^4^ to 10^0^ copies/µL. (**e**) Detection limit determination for G5 with nucleic acid copy numbers ranging from 10^4^ to 10^0^ copies/µL. (**f**) Detection limit determination for G9 with nucleic acid copy numbers ranging from 10^4^ to 10^0^ copies/µL. The levels of statistical significance are denoted as follows: **** *p* < 0.0001. NTC refers to non-target control, and the error bars represent mean ± SD.

**Figure 6 ijms-26-06846-f006:**
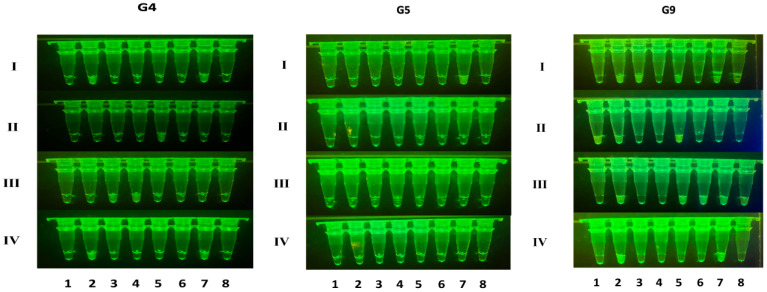
Detection of clinical samples. G4, G5, and G9 infections were detected in 31 clinical samples from pigs using the RT-RAA–CRISPR/cas12a single-tube assay, and the results were visualized by fluorescent tubes.

**Table 1 ijms-26-06846-t001:** Nucleic acid sequences used in this study.

Name	Sequences (5′-3′)	Product Length
RT-RAA-1F	ATGTATGGTATTGAATATACCACAGTTCT	510 bp
RT-RAA-1R	TATATCCATTGGATTACATAACCATTCATT	
RT-RAA-2F	ATGTATGGTATTGAATATACCACAGTTCT	545 bp
RT-RAA-2R	TTCGCTTCATCTGTTTGCTGATAATAATA	
RT-RAA-3F	ATGTATGGTATTGAATATACCACAGTTCT	564 bp
RT-RAA-3R	TCCCATCGATATCCATTTATTCGCTTC	
RT-RAA-4F	ATGTATGGTATTGAATATACCACAGTTCT	476 bp
RT-RAA-4R	ATCAAATCAGCCAATTCAGACATATCTAGCT	
crRNA-G4	UAAUUUCUACUCUUGUAGAUGAUAUCUUUCGUUCUUGUGAG	
crRNA-G5	UAAUUUCUACUCUUGUAGAUCUGUAGAACCACAAUUAUA	
crRNA-G9	UAAUUUCUACUCUUGUAGAUUUAAAACACAAAAUUAUGGA	
FQ-ssDNA reporter	5′-FAM-TTTTTTTTTTT-BHQ-1′	
RT-qPCR-F-G4	CCTATGCTAATTCAACGCAAAATG	102 bp
RT-qPCR-R-G4	GGCCATCCTTTAGTTAGAAACAGTT	
RT-qPCR-F-G5	AGTTACTAGAACAATGGACTTTATC	157 bp
RT-qPCR-R-G5	CAAAAATGTTTCACTTGTAGTTGA	
RT-qPCR-F-G9	TTGATGTAACTACAAGTACCTGTACAA	139 bp
RT-qPCR-R-G9	TATCTAACACTTCTGAGCCACC	

## Data Availability

All the data are available in the manuscript.

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
