# Peer review of "CRISPR/Cas12a-Based One-Tube RT-RAA Assay for PoRV Genotyping"

_ijms, 2025, doi:10.3390/ijms26146846_

Round 1

Reviewer 1 Report

Comments and Suggestions for Authors

The article titled  "CRISPR/Cas12a-Based One-Tube RT-RAA Assay for PoRV Genotyping"  presents a rapid, sensitive, and specific diagnostic method for detecting Porcine Rotavirus in piglets. The authors have used the knowledge of CRISPR/Cas12a detection system and Reverse Transcription-Recombinase-Aided Amplification (RT-RAA) techniques to detect the virus.

-- In the abstract, lines 20 and 21, the authors have mentioned the simultaneous amplification of G4/G5/G9. Please mention if G3 was amplified together or separately.
-- In line 50, it will be ideal to change maturation to evolved.
-- In Figure 1, please mention how were the Cas12a system loaded or adhered to the centrifuge tube cap.
-- ALso mention how the linker between fluorophores gets cleaved with the CRISPR Cas12 system. please describe the mechanism.
-- FOr Figure 2, please give the interpretation of the experiment result.
-- In Figure 2, explain why a faint band is seen in lane 4 from left.
-- It will be ideal if the supplementary figure S4 is brought to main document.
-- In supplementary Figure S5 please mention what is NTC.
-- The mechanism of RT-RAA is not explained n the manuscript.

As such, the article has good experimental results but not well presented. Hence I suggest the authors to reframe the sentences such that it will give an easy grasp to readers.

Author Response

1.In the abstract, lines 20 and 21, the authors have mentioned the simultaneous amplification of G4/G5/G9. Please mention if G3 was amplified together or separately.

Response: Thank you very much for your question.We sincerely appreciate your insightful question. In the first amplification step, the universal primers were designed to amplify all rotavirus subtypes. Subsequently, during the CRISPR-Cas12a system detection step, the G4, G5, and G9 subtypes can be specifically identified and differentiated through our established detection system.

2.In line 50, it will be ideal to change maturation to evolved.

Response: Thank you very much for your suggestion. We have modified the descriptions as suggested in the revised manuscript.

3.In Figure 1, please mention how were the Cas12a system loaded or adhered to the centrifuge tube cap.

Response:Thank you for raising this important question. In our single-tube detection system,“The RT-RAA reaction mixture was first added to the bottom of a centrifuge tube. Subsequently, 10 μL of the Cas12a detection system was loaded onto the inner surface of the tube lid. After gently sealing the lid, the RT-RAA amplification was allowed to proceed to completion. Finally, a brief centrifugation step was performed to transfer the Cas12a system from the lid to the reaction mixture at the bottom of the tube, enabling Cas12a-mediated target cleavage. ”

This design eliminates the need for physical fixation of the Cas12a system due to two key factors: (1) The small volume (10μL) and surface tension of the Cas12a solution allow it to remain stably adhered to the lid under controlled handling conditions, and (2) the closed-tube workflow inherently prevents cross-contamination. We have revised the Methods section (Lines 166-172) to clarify these operational details and emphasize the contamination-control advantages of this integrated approach.

4.ALso mention how the linker between fluorophores gets cleaved with the CRISPR Cas12 system. please describe the mechanism.

Response:Thank you for raising this important question.“An ssDNA reporter molecule labeled with FAM and BHQ-1 was employed for fluorescence detection. Upon activation through specific cleavage of PoRV genomic dsDNA, Cas12a subsequently cleaves the ssDNA reporter molecule via its nonspecific endonuclease activity.”We have made revision according to the recommendations in the revised version.(Line 329-332)

5-FOr figure 2, please give the interpretation of the experiment result.

Response: Thank you for your professional advice. We have supplemented the relevant content as suggested in the revised manuscript.(Line 108-126)

6.In Figure 2, explain why a faint band is seen in lane 4 from left.

Response: Thank you for your professional advice. We sincerely appreciate your insightful feedback and apologize for this oversight in our initial submission. Following your suggestion, we have rigorously re-examined the experimental data and repeated the relevant validation assays. The revised results have been carefully incorporated into the updated manuscript.

7.It will be ideal if the supplementary figure S4 is brought to main document.

Response: Thank you very much for your suggestion. We have modified this as suggested in the revised manuscript.(Line 137)

8.The mechanism of RT-RAA is not explained in the manuscript.

Response: Thank you very much for your suggestion. We have supplemented the relevant content as suggested in the revised manuscript.(Line 46-66)

Reviewer 2 Report

Comments and Suggestions for Authors

This study presents a novel diagnostic platform combining reverse transcription recombinase-aided amplification (RT-RAA) with CRISPR-Cas12a for rapid genotyping of porcine rotavirus (PoRV). The authors address a critical gap in current PoRV surveillance by enabling simultaneous detection of predominant G4/G5/G9 genotypes within 50 minutes at 37°C, achieving 100 copies/μL sensitivity and high concordance with RT-qPCR in clinical validation. However, the manuscript contains some minor points in the presentation and description of the results, as follows:

Revisions:

  1. Line 37: Replace “[3][4]” with “[3,4]”.
  2. Line 42-43 “The limited cross-immunoprotective efficacy between distinct genotypes compromises the comprehensive protective capacity of current vaccines” could be revised to “The limited cross-immunoprotective efficacy among distinct genotypes compromises the comprehensive protective capacity of current vaccines”.
  3. Line 53: Replace “[15][16][17]” with “[15-17]”.
  4. Line 48: The term “Porcine rotavirus (PoRV)” does not appear for the first time in the text. To simplify the writing, it is recommended to write the abbreviation directly writing, it is recommended to choose one of the full name and abbreviation.
  5. Line 54 and 55: We have noticed that “RT qPCR and RT LAMP” appear for the first time in the article. To ensure scientific rigor, please provide the full names.
  6. Line 65: Replace “G5, G9” with "G5 and G9" in this sentence.
  7. Line 69: The “copy” here should be in plural form, replacing “copy” with “copies”.
  8. Line185: The first letter of “specificity” in this line should be capitalized.
  9. Section on "clinical sample detection”. Please clarify whether the research protocol for testing samples was submitted to the Animal Ethics Committee and obtained its approval (IRB approval letter number and date must be provided).
  10. Line219: Replace “42G” with “42 G”.
  11. Line235: Replace "concentration-dependent specificity" with "concentration-dependent activity".
  12. Line247-248: Replace “engineered” with “developed”.
  13. Some of the figures and tables could be improved in terms of clarity and presentation. For example, the labels and annotations in Figure 1 could be made more legible to enhance the readers understanding.
  14. Literature Review: The manuscript could benefit from a more comprehensive review of the recent literature on PoRV detection methods. This would provide a better context for the current study and highlight its contributions to the field.
  15. Detailed Methods: While the methods section is generally well-described, some steps could be elaborated further for clarity. For example, the optimization of the RT-RAA conditions could include more details on the rationale behind the chosen parameters.
  16. Use the correct forms of Latin expressions and their abbreviations, such as "et al." (with a single dot) and "e.g." (for example).
  17. Maintain consistency in the use of tenses. For instance, the Methods and Results sections are typically written in the past tense, while the Introduction is often in the present tense.
  18. Line312: Replace “RTRAA” with “RT-RAA”.

Author Response

Revisions:

1.Line 37: Replace “[3][4]” with “[3,4]”.

Response: Thank you for your suggestion. We have modified the descriptions as suggested.

2.Line 42-43 “The limited cross-immunoprotective efficacy between distinct genotypes

compromises the comprehensive protective capacity of current vaccines” could be revised to “The limited cross-immunoprotective efficacy among distinct genotypes compromises the comprehensive protective capacity of current vaccines”

Response: Thank you for your suggestion. We have modified the descriptions as recommended.

3.Line 53: Replace “[15][16][17]” with “[15-17]”.

Response: Thank you for your suggestion. We have modified the descriptions as recommended.

4.Line 48: The term “Porcine rotavirus (PoRV)” does not appear for the first time in the text. To

simplify the writing, it is recommended to write the abbreviation directly writing, it is recommended to choose one of the full name and abbreviation.

Response: Thank you for your suggestion. We have modified the descriptions as recommended.

5.Line 54 and 55: We have noticed that “RT qPCR and RT LAMP” appear for the first time in the

article. To ensure scientific rigor, please provide the full names.

Response: Thank you for your suggestion. We have modified the descriptions as recommended.

6.Line 65: Replace “G5, G9” with "G5 and G9" in this sentence.

Response: Thank you for your suggestion. We have modified the descriptions as recommended.

7.Line 69: The “copy” here should be in plural form, replacing “copy” with “copies”.

Response: Thank you for your suggestion. We have modified the descriptions as recommended.

8.Line185: The first letter of “specificity” in this line should be capitalized.

Response: Thank you for your suggestion. We have added the relevant description as recommended in the revised manuscript.

9.Section on "clinical sample detection”. Please clarify whether the research protocol for testing

samples was submitted to the Animal Ethics Committee and obtained its approval (IRB approval letter number and date must be provided).

Response: Thank you for reviewing our manuscript and providing valuable feedback. Below we address your inquiries regarding sample sources and ethical review documentation:

Sample Collection Protocol
The porcine samples were collected from multiple local pig farms. For pigs exhibiting diarrheal symptoms, farm veterinarians performed routine anal swab sampling as part of standard diagnostic testing. After completing these clinical tests, residual samples were preserved for our study. We fully informed farm owners about the research purpose of sample collection and obtained their explicit consent.

Ethical Compliance
All samples were obtained during routine veterinary inspections at the farms. In accordance with Section 6.7.1 of the Chinese National Standard GB/T 35892-2018 (Guidelines for Ethical Review of Animal Welfare in Laboratory Animals, ICS 65.020.30; CCS B44), priority is given to alternative methods that avoid unnecessary harm to animals. Following consultation with our Institutional Animal Welfare and Ethics Committee, it was determined that:

Samples were residual materials from standard diagnostic/therapeutic procedures

No additional invasive interventions were performed

No sensitive personal data or commercial interests were involved
Thus, no separate ethical approval was required. This approach aligns with the 3Rs principles (Replacement, Reduction, Refinement) governing humane animal research.

We rigorously adhere to ethical standards and ensure full compliance with all applicable regulations. Thank you again for your insightful comments and guidance.

10.Line219: Replace “42G” with “42 G”.

Response: Thank you for your suggestion. We have modified the descriptions as recommended.

11.Line235: Replace "concentration-dependent specificity" with "concentration-dependent activity".

Response: Thank you for your suggestion. We have modified the descriptions as recommended.

12.Line247-248: Replace “engineered” with “developed”.

Response: Thank you for your suggestion. We have modified the descriptions as recommended.

13.Some of the figures and tables could be improved in terms of clarity and presentation. For

example, the labels and annotations in Figure 1 could be made more legible to enhance the readers understanding.

Response: Thank you for your suggestion. We have revised the relevant figures as recommended in the manuscript.

14.Literature Review: The manuscript could benefit from a more comprehensive review of the

recent literature on PoRV detection methods. This would provide a better context for the current study and highlight its contributions to the field.

Response: Thank you for pointing out this issue. We have added the information as suggested in the revised manuscript. Line 69-79.

15.Detailed Methods: While the methods section is generally well-described, some steps could be

elaborated further for clarity. For example, the optimization of the RT-RAA conditions could include more details on the rationale behind the chosen parameters. 

Response: Thank you for your suggestion. We have revised the relevant description as recommended in the manuscript.Line 162-168.

16.Use the correct forms of Latin expressions and their abbreviations, such as "et al." (with a single

dot) and "e.g." (for example).

Response: Thank you for your suggestion. We have revised the relevant description as recommended in the manuscript.

17.Maintain consistency in the use of tenses. For instance, the Methods and Results sections are

typically written in the past tense, while the Introduction is often in the present tense.

Response: Thank you for your suggestion. We have revised the manuscript as suggested.

18.Line312: Replace “RTRAA” with “RT-RAA”.

Response: Thank you for your suggestion. We have revised the manuscript as suggested.

Round 2

Reviewer 1 Report

Comments and Suggestions for Authors

Please cite if there is any reference for adherence of proteins on lid by surface tension. Else please give the detailed mechanism. Are the results same when the Cas12a was added separately or when adhered to the lid by surface tension?

Author Response

Thank you for your insightful question. Small droplets adhere to the tube lid through surface tension. Proteins dissolved within these droplets enable their attachment via surface tension, as explained in the cited literature[1]. According to the model described in this reference, factors influencing droplet surface tension include wetting and contact angles. The addition of protein-based components has minimal impact on surface tension (slightly increasing viscosity but exerting negligible effects on droplet adherence to the lid in our experimental setup). Another study similarly reported that droplets with various components affected the contact angle of droplets in different ways, while their influence on surface tension remained insignificant. Notably, surface tension at the solid-liquid interface was primarily influenced, whereas it showed little correlation with liquid concentration[2]. Furthermore, numerous published CRISPR-Cas12a-based one-pot detection methods employ approaches analogous to ours. For instance, Reference[3] describes: "A 10 µL/reaction RPA mix and 7 µL/reaction Cas12a mix were respectively loaded on the bottom of the tube and on the lid without opening the lid during detection. Then, 3 µL/reaction sample mix was added into the RPA mix, and the lid was closed quickly." Similar methodologies have been extensively documented[4]. Additionally, comparative studies between two-step detection methods (with Cas12a protein added separately) and one-pot single-step approaches demonstrated equivalent detection sensitivities[5]. These collective findings support the robustness and validity of our experimental design.

  1. Extrand, C. W., Surface tension of very small liquid drops. Colloids and Surfaces a-Physicochemical and Engineering Aspects 2023,679.
  2. Xu, J. Z.; Jia, L.; Dang, C.; Liu, X. Y.; Ding, Y., Effects of solid-liquid interaction and mixture concentration on wettability of nano-droplets: Molecular dynamics simulations. Aip Advances 2022,12, (10).
  3. Xiong, Y.; Cao, G.; Chen, X.; Yang, J.; Shi, M.; Wang, Y.; Nie, F.; Huo, D.; Hou, C., One-pot platform for rapid detecting virus utilizing recombinase polymerase amplification and CRISPR/Cas12a. Appl Microbiol Biotechnol 2022,106, (12), 4607-4616.
  4. Ma, L.; Zhu, M.; Meng, Q.; Wang, Y.; Wang, X., Real-time detection of Seneca Valley virus by one-tube RPA-CRISPR/Cas12a assay. Front Cell Infect Microbiol 2023,13, 1305222.
  5. Zeng, X.; Jiang, Q.; Yang, F.; Wu, Q.; Lyu, T.; Zhang, Q.; Wang, J.; Li, F.; Xu, D., Establishment and optimization of a system for the detection of Candida albicans based on enzymatic recombinase amplification and CRISPR/Cas12a system. Microbiol Spectr 2025,13, (5), e0026825.